# Scalable Multimodal Variational Autoencoders with Surrogate Joint Posterior

## Abstract

To obtain a joint representation from multimodal data in variational autoencoders (VAEs), it is important to infer the representation from arbitrary subsets of modalities after learning. A scalable way to achieve this is to aggregate the inferences of each modality as experts. A state-of-the-art approach to learning this aggregation of experts is to encourage all modalities to be reconstructed and cross-generated from arbitrary subsets. However, this learning may be insufficient if cross-generation is difficult. Furthermore, to evaluate its objective function, exponential generation paths concerning the number of modalities are required. To alleviate these problems, we propose to explicitly minimize the divergence between inferences from arbitrary subsets and the surrogate joint posterior that approximates the true joint posterior. We also proposed using a gradient origin network, a deep generative model that learns inferences without using an inference network, thereby reducing the need for additional parameters by introducing the surrogate posterior. We demonstrate that our method performs better than existing scalable multimodal VAEs in inference and generation.

## 1 Introduction

Our world is characterized by many kinds of information. This *multimodal* information can be used to form common concepts. For instance, by recognizing information, such as the landscape of the beach and the salty scent of the ocean, we can form a concept about the sea. In multimodal learning (Baltrušaitis et al., 2018), it has been studied as the learning of joint representations from multimodal data, especially using variational autoencoders (VAEs) (Kingma & Welling, 2013; Rezende et al., 2014).

The goal of multimodal representation learning with VAEs (Suzuki et al., 2016; Vedantam et al., 2018; Wu & Goodman, 2018; Shi et al., 2019) is to learn all given multimodal data and to infer a joint representation from arbitrary subsets of modalities. This representation is useful as an input to predict the corresponding label information or as an intermediate representation to generate other modalities, i.e., cross-generations. However, this is difficult because a joint representation must be inferred even when some of the modalities are missing at test time. The most naive way to deal with this difficulty is to have inference models for all possible combinations of modalities, which is exponentially necessary given the number of modalities.

A scalable way to address this is to consider the inferences for each modality as experts and combine them to approximate the inferences from an arbitrary subset of modalities. There are two types of methods for aggregating inferences: products of experts (PoE) (Hinton, 2002) and mixtures of experts (MoE); VAEs that use these as approximate posteriors are proposed as MVAE (Wu & Goodman, 2018) and MMVAE (Shi et al., 2019), respectively. Each method has different advantages that exist in a trade-off relationship, i.e., PoE increases the amount of mutual information with joint representation by increasing the number of modalities, while MoE has the effect of encouraging cross-generation between modalities.

Sutter et al. (2021) defined the objective of multimodal VAEs to be that the inference of all possible subsets approximate the true joint posterior, and derived a generalized evidence lower bound (ELBO) as a lower bound of that objective. This ELBO can be viewed as using a mixture of products of experts (MoPoE) as an aggregation method for inference, which is to generate all modalities from PoE-aggregated inference of arbitrary subsets. In other words, this model, called MoPoE-VAE,

tries to achieve the above objective of multimodal VAEs by minimizing all reconstruction and cross-generation errors in all modality combinations. They showed that this is a generalization that has the advantages of both PoE and MoE. However, these properties of MoPoE-VAE imply that inference may not be sufficiently learned when cross-generation is difficult. Furthermore, the number of paths for this reconstruction and cross-generation in MoPoE-VAE is required to be exponential with respect to the number of modalities. Therefore, we pursue a scalable method that realizes the above objective of multimodal VAEs.

In this paper, we propose a different approach to achieve this objective. We first prepare a surrogate joint posterior, which approximates the true joint posterior given all modalities, and then minimize the divergence between the PoE-aggregated inferences and this surrogate posterior. The approximation of surrogate posterior is perform by the usual VAE optimization, and we simultaneously perform this optimization and the divergence minimization above.

The advantages of this new approach are twofold. First, if the surrogate model is sufficiently close to the true posterior, then optimizing this objective function explicitly achieves the objective of multimodal VAEs defined in MoPoE, i.e, approximating inferences from arbitrary subsets to the true posterior. Second, unlike MoPoE, this method is not exponentially more expensive. This is because this method is based on divergence minimization of all inference models, rather than reconstruction or cross-generation of all modalities in combination. Although this method requires an additional surrogate model, the size is not exponential as in previous similar methods (Suzuki et al., 2016; Vedantam et al., 2018), but only doubles the number of parameters in encoders. Therefore, we call this method a *scalable multimodal VAE (SMVAE)*.

Furthermore, to reduce the number of parameters in the surrogate posterior, we propose to use variational gradient origin networks (VGONs) (Bond-Taylor & Willcocks, 2020) for training the surrogate posterior. VGONs are deep generative models that can rapidly acquire amortized inference to representations without a trainable network. We call our method that uses VGONs for surrogate posterior SMVGON.

Experiments show that the proposed method performs better than conventional methods in inferring joint representations and generating modalities on commonly used multimodal benchmarks. In particular, we show that the proposed method improves the performance in situations where the conventional methods have difficulty in cross-generation. We also confirm that the performance of SMVGONs with VGONs is comparable to that of SMVAEs when the input modalities are relatively simple, without the need for additional inference networks for surrogate posteriors.

## 2 PRELIMINARY

### 2.1 MULTIMODAL VAES

Suppose that we are given an i.i.d. dataset $\{X^{(i)}\}_{i=1}^{N}$, where each example is a set of $M$ modalities $X^{(i)} = \{\mathbf{x}_m^{(i)}\}_{m=1}^{M}$ and where each modality $\mathbf{x}_m^{(i)} = [x_{1m}^{(i)}, ..., x_{D_m m}^{(i)}]$ has its own dimension $D_m$ and domain. We assume that these examples are derived from data distribution $p_{data}(X)$ and that each example $X^{(i)}$ has a corresponding common latent concept $\mathbf{z}^{(i)} = [z_1^{(i)}, ..., z_j^{(i)}, ..., z_J^{(i)}] \in \mathbb{R}^J$, i.e., a joint representation.

Given a training set $\{X^{(i)}\}_{i=1}^{N}$, our objective is to infer a joint representation from the subset of modalities[1] $X_s \subseteq X$ and to generate another subset $X_s' \subseteq X$ via that representation. In other words, we aim to obtain an inference $p(\mathbf{z}|X)$ and a joint distribution $p(X, \mathbf{z}) = \prod_{\mathbf{x}_m \in X} p(\mathbf{x}_m|\mathbf{z})p(\mathbf{z})$ using a training set. In this study, we designate the generation when $X_s = X_s'$ as *reconstruction* and when $X_s \cap X_s' = \emptyset$ as *cross-generation*.

To achieve this, we use VAEs (Kingma & Welling, 2013; Rezende et al., 2014), which are deep generative models that can learn inference in addition to generative models. Therefore, we can acquire representations of data by inference. The generative model $p_{\Theta}(X|\mathbf{z}) = \prod_{\mathbf{x}_m \in X} p_{\theta_m}(\mathbf{x}_m|\mathbf{z})$, where $\Theta = \{\theta_m\}_{m=1}^{M}$ is a set of parameters, is parameterized by deep neural networks, and the prior of

---

[1]We denote the multimodal set and any subset of it by $X$ and $X_s$, respectively. We also omit the index of each example for brevity.

the latent variable is set as a standard Gaussian $p(\mathbf{z}) = \mathcal{N}(\mathbf{0}, \mathbf{I})$. The objective function of VAEs is to maximize the marginal log-likelihood of the multimodal dataset: $\sum_{i=1}^{N} \log p_\Theta(X_i)$. However, since it is tractable to optimize this likelihood directly, we introduce an inference model $q_\Phi(\mathbf{z}|X)$ to approximate the posterior $p_\Theta(\mathbf{z}|X)$, and instead optimize the following lower bound of the marginal log-likelihood given $X$, which is called the ELBO:

$$\mathcal{L}(\Theta, \Phi; X) \equiv \mathbb{E}_{q_\Phi(\mathbf{z}|X)}[\log p_\Theta(X|\mathbf{z})] - D_{KL}(q_\Phi(\mathbf{z}|X)||p(\mathbf{z})). \tag{1}$$

Here, the optimization of ELBO with respect to the parameters of the inference model corresponds to minimizing the following Kullback-Leiber (KL) divergence:

$$D_{KL}(q_\Phi(\mathbf{z}|X)||p_\Theta(\mathbf{z}|X)). \tag{2}$$

In other words, maximizing the ELBO encourages bringing the true posterior closer to the inference model.

After optimizing Equation 1, we aim to infer a joint representation $\mathbf{z}$ from an arbitrary subset of modalities $X_s$. However, this is challenging because Equation 1 only models joint inferences from all modalities. Therefore, for inferences from a subset of modalities $X_s$, other modality inputs are missing, which results in the inability to infer the representation properly (Suzuki et al., 2016).

A scalable way to alleviate this difficulty is to consider the inference model for each modality as an *expert* and approximate the inference model for arbitrary modalities by combining them. There are two ways to combine experts: product and mixture.

### 2.1.1 Aggregated inference with PoE and MoE

A joint inference given all modalities $X$ is expressed as the following using PoE (Hinton, 2002):

$$q_{PoE}(\mathbf{z}|X) \propto p(\mathbf{z}) \prod_{\mathbf{x}_m \in X} q_{\phi_m}(\mathbf{z}|\mathbf{x}_m). \tag{3}$$

In general, PoE cannot be calculated as closed-form, but if we restrict each expert to Gaussian $q_{\phi_m}(\mathbf{z}|\mathbf{x}_m) = \mathcal{N}\left(\boldsymbol{\mu}_m, \operatorname{diag}(\boldsymbol{\sigma}_m^2)\right)$, then Equation 3 is calculable as $q_{PoE}(\mathbf{z}|X) = \mathcal{N}(\boldsymbol{\mu}, \operatorname{diag}(\boldsymbol{\sigma}^2))$, where $\mu_j = \frac{\sum_m \mu_{jm} \sigma_{jm}^{-2}}{\sum_m \sigma_{jm}^{-2}}$ and $\sigma_j^2 = \frac{1}{\sum_m \sigma_{jm}^{-2}}$. Therefore, the Gaussian PoE enables us to infer the representation from subset $X_s$.

However, if we optimize this inference model with Equation 1, then the inference of arbitrary modalities is not optimized explicitly. Wu & Goodman (2018) proposed to train ELBOs with PoE-aggregated inferences of all modality combinations $\sum_{X_s \in \mathcal{X}} \mathcal{L}(\Theta_s, \Phi_s; X_s)$; however, since the number of ELBOs increases exponentially, they used the subsampled version. The model that learns PoE inference by this objective function is called MVAE (Wu & Goodman, 2018).

Another method to combine experts is to use MoE:

$$q_{MoE}(\mathbf{z}|X) = \sum_{\mathbf{x}_m \in X} w_m q_{\phi_m}(\mathbf{z}|\mathbf{x}_m), \tag{4}$$

where $w_m$ is a mixing coefficient of each expert and where $\sum_{\mathbf{x}_m \in X} w_m = 1$. Shi et al. (2019) proposed a VAE with MoE-aggregated inference models called MMVAE.

There is a trade-off between the approximate posterior with PoE and MoE. PoE can perform inference given any subset and MVAE can learn inference explicitly during training. PoE also has the property that increasing the size of the subset, i.e., the number of input modalities, leads to an increase in the information content of the inferred latent variables. However, the objective function of MVAE does not guarantee that the inferences from any subset are the same, since it only learns reconstruction and not cross-generation. Sutter et al. (2021) also pointed out that its objective function by subsampling is not a valid lower bound.

MMVAE based on MoE, on the other hand, has a valid ELBO and learns to explicitly encourage cross-generation, which ensures that the inferences from each modality remain consistent. However, this method only learns inferences from a single modality, and increasing the number of modalities does not necessarily increase the information content of the representation.

### 2.1.2 MoPoE-VAE

Sutter et al. (2021) considered optimization on all possible subsets, i.e., the powerset $\mathcal{P}(X)$, and reestablished the minimization of the following KL divergence as the objective of multimodal VAEs:

$$\arg\min_{\phi} \sum_{X_c \in \mathcal{P}(X)} D_{\mathrm{KL}}\left(q_{\phi_c}\left(\mathbf{z}|X_k\right) \| p_{\Theta}(\mathbf{z}|X)\right). \tag{5}$$

They then proposed a generalized ELBO that is a valid lower bound on the log-likelihood $p_{\Theta}(X)$ and whose maximization corresponds to the minimization of Equation 5:

$$\mathbb{E}_{q_{MoPoE}(\mathbf{z}|X)}[\log p_{\Theta}(X|\mathbf{z})] - D_{KL}(q_{MoPoE}(\mathbf{z}|X)\|p(\mathbf{z})), \tag{6}$$

where the form of the posterior of $q_{MoPoE}(\mathbf{z}|X)$ is called MoPoE and is defined as follows:

$$q_{MoPoE}(\mathbf{z}|X) = \frac{1}{2^M} \sum_{X_c \in \mathcal{P}(X)} q_{PoE}(\mathbf{z}|X_c). \tag{7}$$

A feature of the model that maximizes Equation 6, called MoPoE-VAE, is that it learns to generate all modalities from all possible subsets, including reconstruction and cross-generation, which improves the performance of inference and generation from all subsets. In addition, they showed that this MoPoE-VAE is a generalization of MVAE and MMVAE, specifically MVAE when the summation in Equation 7 is taken only for a set of all modalities $X$ and MMVAE when it is taken for the single modalities $\{\mathbf{x}_m\}_{m=1}^M$. Therefore, MoPoE-VAE can be regarded as having the advantages of both MVAE and MMVAE.

### 2.2 Variational Gradient Origin Networks

VGONs (Bond-Taylor & Willcocks, 2020) are deep generative models that can perform amortized inference equivalent to VAEs without an inference network.[2]

Given an observed variable $\mathbf{x}$ and a latent variable $\mathbf{z}$, let the generative model be $p_{\theta}(\mathbf{x}|\mathbf{z})$, and the inference model be $q_{\phi,\mathbf{z}_0}(\mathbf{z}) = \mathcal{N}(\mu_{\phi}(\mathbf{z}_0), \sigma_{\phi}^2(\mathbf{z}_0)\mathbf{I})$ with the mean and variance mapped by trainable linear transformations ($\mu_{\phi}$ and $\sigma_{\phi}^2$) from a parameter $\mathbf{z}_0$. This objective function, ELBO, given $\mathbf{x}$ is defined as follows:

$$\mathcal{L}_{VGON}(\theta, \phi, \mathbf{z}_0; \mathbf{x}) \equiv \mathbb{E}_{q_{\phi,\mathbf{z}_0}(\mathbf{z})}[\log p_{\theta}(\mathbf{x}|\mathbf{z})] - D_{KL}(q_{\phi,\mathbf{z}_0}(\mathbf{z})\|p(\mathbf{z})). \tag{8}$$

In VGONs, the estimator for $\mathbf{z}_0$ is calculated by a single update based on the gradient of Equation 8:

$$\hat{\mathbf{z}}_0 = \mathbf{z}_0 + \nabla_{\mathbf{z}_0}\mathcal{L}_{VGON}(\theta, \phi, \mathbf{z}_0; \mathbf{x}), \tag{9}$$

where $\mathbf{z}_0 = \mathbf{0}$, i.e, the initial value of $\mathbf{z}_0$ is set to the origin. Using this estimator $\hat{\mathbf{z}}_0$, we can obtain inference as $\mathbf{z} \sim q_{\phi,\hat{\mathbf{z}}_0}(\mathbf{z})$ and can evaluate the objective function as $\mathcal{L}_{VGON}(\theta, \phi, \hat{\mathbf{z}}_0; \mathbf{x})$. In learning, we also backpropagate the gradient of the objective function through Equation 9, which allowed us to learn both inference and generation simultaneously as in VAEs.

## 3 Methods

### 3.1 Issues of MoPoE-VAE

MoPoE-VAE aims at the objective of Equation 5 by maximizing the ELBO of Equation 6. The two are equivalent in variational inference, but have different implications for learning including generative models, i.e., whether the Equation 5 is achieved depends on the performance of all modalities of reconstruction and cross-generation. Therefore, in situations where cross-generation is difficult, this objective might not be fully achieved, specifically, the inference and generation of some modalities might be significantly worse than others.

---

[2]In the original paper (Bond-Taylor & Willcocks, 2020), gradient origin networks were introduced first, and then variational GONs and implicit GONs were introduced as an extension of them.

Another issue is that the computational cost of evaluating Equation 6 increases exponentially with the number of modalities. Specifically, we consider a computational path that generates modalities from inferred representations. In this case, since there are $2^M$ representations to be inferred by MoPoE, the generation path required to evaluate Equation 6 is $M \times 2^M$; therefore, as the number of modalities increases, the evaluation of the ELBO becomes more intractable and might also lead to insufficient learning of cross-generation.

## 3.2 A SURROGATE JOINT POSTERIOR AND THE REFORMULATION OF THE OBJECTIVE

To address the above issues, we aim to explicitly realize Equation 5. However, since a true posterior cannot be obtained, we prepare a *surrogate joint posterior* $q_\Lambda(\mathbf{z}|X)$, which is a Gaussian distribution parameterized by a neural network and should approximate the true posterior $p_\Theta(\mathbf{z}|X)$. We then reformulate our objective as minimizing the KL divergence between subset inferences and this surrogate posterior:

$$\arg\min_\phi \sum_{X_c \in \mathcal{P}(X)} D_{\mathrm{KL}}\left(q_{\phi_c}\left(\mathbf{z}|X_c\right)\|q_\Lambda(\mathbf{z}|X)\right). \tag{10}$$

Here, we consider each inference from a subset $q_{\phi_c}\left(\mathbf{z}|X_c\right)$ to be expressed in PoE form. Then, since the surrogate posterior is defined as Gaussian, this KL divergence can be calculated analytically.

If $D_{KL}(q_\Lambda(\mathbf{z}|X)\|p_\Theta(\mathbf{z}|X)) \approx 0$ is satisfied, then this optimization problem is the same as Equation 5. As explained in Section 2.1, minimizing the KL divergence between the approximate posterior (in this case, the surrogate posterior) and the true posterior is equivalent to maximizing the ELBO. Therefore, we can first maximize the ELBO $\mathcal{L}(\Theta, \Lambda; X)$ sufficiently, and then fix the surrogate posterior and perform the optimization of Equation 10. However, this requires a two-step learning process, which can be time consuming. Therefore, we propose to optimize these two steps simultaneously, referring to existing multimodal VAEs with additional inference models such as JMVAE (Suzuki et al., 2016) (see Section 4), i.e., maximize the following objective function:

$$\mathcal{L}(\Theta, \Lambda; X) - \frac{\beta}{2^M} \sum_{X_c \in \mathcal{P}(X)} D_{\mathrm{KL}}\left(q_{\phi_c}\left(\mathbf{z}|X_c\right)\|q_\Lambda(\mathbf{z}|X)\right), \tag{11}$$

where $\beta$ is a hyperparameter that adjusts for the effect of KL divergence in the objective function. We train this objective function in an end-to-end fashion. Empirically, we have found that this end-to-end optimization results in higher performance than the two-stage optimization (see Appendix D). At test time, we infer representations from subsets using a PoE-aggregated inference model $q_{\phi_c}\left(\mathbf{z}|X_c\right)$.

The advantages of this objective function are as follows. First, inference models from arbitrary subsets are learned not by cross-generation, but by explicitly minimizing the divergence between them and the surrogate posterior. Moreover, the surrogate posterior is learned based on the maximization of ELBO $\mathcal{L}(\Theta, \Lambda; X)$, i.e., based on the reconstruction in all modalities, rather than cross-generation. Therefore, this objective can mitigate the impact of the difficulty of cross-generation on the performance of inference and generation.

Secondly, the computational cost can be significantly reduced compared to MoPoE-VAE. There is only one pass for generation in evaluating this objective function for each modality in $\mathcal{L}(\Theta, \Lambda; X)$. In addition, the second term of Equation 11 computes the inference for all possible subsets, which is obtained by aggregating the inferences for each modality; therefore, the inference pass is also only once for each modality.

We call the proposed model a *scalable multimodal VAE (SMVAE)* because it is scalable to the number of modalities while explicitly optimizing toward the objective of multimodal VAEs.

## 3.3 REDUCING THE PARAMETERS IN SURROGATE INFERENCE MODELS

Although the proposed method alleviate the issues of previous multimodal VAEs, it also has the disadvantage of requiring more parameters for the inference model. The added surrogate posterior requires as many parameters as all of the unimodal inference models, i.e., the parameter size of the inference model is twice as large as that of previous models such as MoPoE-VAE. Note that in existing studies of multimodal VAEs with additional inference models, such as JMVAE, the parameter

size increases exponentially because inference models are required for each subset. In comparison, SMVAE keeps the number of required parameters linear with respect to the number of modalities.

To reduce the number of required parameters, we propose to use VGONs to train the surrogate posterior and generative models for all modalities. In the VGON framework, the surrogate posterior given multimodal inputs can be defined as $q_{\lambda, \mathbf{z}_0}(\mathbf{z}) = \mathcal{N}(\mu_\lambda(\mathbf{z}_0), \sigma_\lambda^2(\mathbf{z}_0)\mathbf{I})$, where the estimator of $\mathbf{z}_0$ is calculated as $\hat{\mathbf{z}}_0 = \nabla_{\mathbf{z}_0} \mathcal{L}_{VGON}(\Theta, \lambda, \mathbf{z}_0; X)$. Therefore, the objective function of the proposed method using VGONs is as follows:

$$\mathcal{L}_{VGON}(\Theta, \lambda, \mathbf{z}_0; X) - \frac{\beta}{2^M} \sum_{X_c \in \mathcal{P}(X)} D_{\mathrm{KL}}\left(q_{\phi_c}(\mathbf{z}|X_c) \| q_{\lambda, \mathbf{z}_0}(\mathbf{z})\right). \tag{12}$$

We refer to this model as *SMVGON* to distinguish it from SMVAE.

Note that the surrogate posterior is used to approximate the aggregated inference model during training and is not used during testing; therefore, it is possible to use other iterative inference algorithms. In this study, however, we used VGONs, which can perform inference in a single iteration, to avoid slow training.

## 4 RELATED WORKS

For learning unsupervised joint representations from multimodal data, autoencoder-based methods have been used (Ngiam et al., 2011; Silberer & Lapata, 2014); however, they have difficulties in the challenge of missing modalities. Joint representation learning using deep belief networks (Hinton, 2009) or deep Boltzmann machines (Salakhutdinov & Hinton, 2009) can adequately handle missing modalities (Srivastava & Salakhutdinov, 2012; Srivastava et al., 2012; Sohn et al., 2014), but has difficulty scaling to large dataset.

Early multimodal learning using VAEs dealt with cross-generation of two modalities, such as conditional VAEs (Sohn et al., 2015; Kingma et al., 2014) and conditional multi-modal autoencoders (Pandey & Dukkipati, 2017). However, because these methods directly learn conditional generative models, they cannot perform bidirectional generation between modalities, nor can they obtain joint latent representations of different modalities.

Some studies of multimodal VAEs require additional inference models (Suzuki et al., 2016; Vedantam et al., 2018; Korthals et al., 2019; Wu & Goodman, 2019). Suzuki et al. (2016) pointed out that VAEs trained on all modalities have the problem of missing modalities in inference, and propose to approximate the unimodal inference models to explicitly approach the joint inference model of VAEs. The objective function of JMVAE is the ELBO of VAE given multimodal inputs minus the KL divergence corresponding to the above approximation.[3] Vedantam et al. (2018) introduced TELBO, where the objective function is composed of the sum of ELBOs that take each set of modalities as input, using the same models as JMVAE. Korthals et al. (2019) introduced M²VAE that can be regarded as a combination of JMVAE and TELBO, and Wu & Goodman (2019) proposed VAEVAE that excludes a KL divergence term between joint inference and prior from it. They are learned end-to-end with a single objective function, except for TELBO.

The challenge with these models is that they require inference models for all modality combinations, i.e., the number of inference models grows exponentially with the number of modalities. Our proposed method is similar to these methods; however, unlike these methods, required memory cost is linear in the number of modalities. Kutuzova et al. (2021) proposed to introduce additional unimodal inference models instead of a joint inference model, and to reduce the number of required inference models to the square of the number of modalities. However, this also does not scale with the number of modalities compared to our linear cost model.

MMJSD proposed by Sutter et al. (2020) is closer to our work, i.e., it encourages unimodal inference to be closer to joint inference from all modalities. The differences between our work and

---

[3]When the number of modalities is two, our proposed method is similar to JMVAE, except that the direction of KL divergence is reversed and PoE-aggregation is used for inference. Please refer to Appendix A for the similarity and performance difference between JMVAE and SMVAE. When the number of modalities is two, the performance of the representations is comparable, but SMVAE outperforms in the cross-generation.

MMJSD are twofold. First, while MMJSD only considers unimodal inference, our work approximates joint inference with any combination of unimodals, which allows to learn inferences from arbitrary modalities. The second is the design of a joint inference. MMJSD considers a dynamic prior that combines all unimodal inferences and considers it a joint inference. However, there is no guarantee that this dynamic prior approximates the true posterior as in the surrogate posterior of SMVAE. In addition, since MMJSD uses the same networks for a joint inference and unimodal inferences, it might be difficult for this joint inference to achieve all of the approximations to the true posterior and to inferences from arbitrary modalities.

The other topic in multimodal VAEs is the acquisition of modality-specific latent variables, apart from the global latent variables that embed all modalities (Tsai et al., 2018; Hsu & Glass, 2018; Sutter et al., 2020). Our study is in a different direction from these, since we are addressing the possibility of inferring a single joint representation from arbitrary subset of modalities. In the experiments, we confirm that SMVAE achieves high performance with a single latent variable in a task where the conventional method achieves by adding modality-specific latent variables.

Recently, contrastive-based multimodal (multi-view) learning (Tian et al., 2019; Alayrac et al., 2020; Tsai et al., 2021) has been gaining attention instead of autoencoder-based, but these aim to acquire *coordinated* representations (Baltrušaitis et al., 2018), i.e., they do not assume inference from a subset of arbitrary modalities.

## 5 EXPERIMENTS

We use MNIST-SVHN-Text (Sutter et al., 2020), Bimodal CelebA (Sutter et al., 2020), and PolyMNIST (Sutter et al., 2021) datasets for evaluation. MNIST-SVHN-Text is the MNIST-SVHN (Shi et al., 2019) with the addition of the text modality, which represents the English name of each number as text and randomly changes the starting index to give a diversity. Bimodal CelebA is a dataset of face images from the CelebA (Liu et al., 2015) with additional text describing the corresponding attribute labels. The PolyMNIST dataset consists of a set of five different MNIST images with different backgrounds and handwriting styles, each of which is considered a different modality. Different modalities of the same example have the same digits.

We confirm that the proposed model realizes the objective of Equation 5, i.e., it can properly infer representations and generate modalities from arbitrary subsets. To evaluate this quantitatively, we estimate the performance of the representation by training a linear classifier under inference from all modalities and predicting by it from the representation inferred from subsets. If this evaluation results in high performance on different subsets, it means that the inference from the subsets is successful. We also perform cross-generation from arbitrary modalities and evaluate the performance of the generated modalities by pre-trained classifiers from Sutter et al. (2020). [4] The interpretation of this evaluation is the same as in the case of representation.

For the network architectures, we follow Sutter et al. (2020), with 32-dimensional latent variables for MNIST-SVHN-Text and Bimodal CelebA and 256 for PolyMNIST. The distributions for images are set to Laplace, and Text is set to categorical. To align the scaling of the likelihoods for each modality, we divided each likelihood by the dimension of each modality and multiplied the largest dimension value by all likelihoods so that the coefficient of the likelihood term with the largest dimension was 1. We use Adam (Kingma & Ba, 2014) for optimization, with a learning rate of 0.001, and a mini-batch size of 256. We set $\beta = 1$, and train 50 epochs for MNIST-SVHN-Text and Bimodal CelebA and 300 epochs for PolyMNIST. All experimental results are the average of three runs. We implement them using PyTorch (Paszke et al., 2019) and Pixyz (Suzuki et al., 2021).

### 5.1 MNIST-SVHN-TEXT

Table 1 shows the results of the performance of representations in MNIST-SVHN-Text. MVAE has better performance for joint representations, but poorer performance for representations inferred from single modalities. MMVAE has better performance for inference from single modalities, but worse performance for joint inference. The performance of MoPoE compensates for the shortcomings of both methods. In contrast, the proposed method outperforms MoPoE in inference from

---

[4]https://github.com/thomassutter/MoPoE

Table 1: Classification accuracy on the latent representation in MNIST-SVHN-Text. Bold represents the highest accuracy in each modality.

| MODEL | M | S | T | M+S | M+T | S+T | M+S+T |
|---|---|---|---|---|---|---|---|
| MVAE | 0.82 | 0.43 | 0.85 | 0.86 | 0.98 | 0.82 | 0.98 |
| MMVAE | **0.97** | **0.83** | **0.99** | 0.90 | **0.99** | 0.91 | 0.93 |
| MoPoE-VAE | 0.88 | 0.75 | **0.99** | 0.91 | 0.98 | 0.93 | 0.98 |
| **SMVAE** | 0.95 | 0.78 | **0.99** | 0.96 | **0.99** | **0.99** | **0.99** |
| **SMVGON** | 0.96 | 0.80 | **0.99** | 0.98 | **0.99** | **0.99** | **0.99** |

Table 2: Classification accuracy of image generations on MNIST-SVHN-Text. The top line shows modalities generated, and the second line shows modalities conditioned when generating it.

| MODEL | M | | | | S | | | | T | | | |
|---|---|---|---|---|---|---|---|---|---|---|---|---|
| | S | T | S+T | M+S+T | M | T | M+T | M+S+T | M | S | M+S | M+S+T |
| MVAE | 0.23 | 0.17 | 0.27 | 0.95 | 0.44 | 0.37 | 0.65 | 0.78 | **0.97** | 0.19 | 0.25 | **0.99** |
| MMVAE | **0.80** | **0.99** | 0.91 | 0.92 | 0.36 | 0.38 | 0.37 | 0.45 | **0.97** | **0.82** | 0.89 | 0.93 |
| MoPoE-VAE | 0.79 | **0.99** | 0.94 | 0.96 | 0.34 | 0.35 | 0.35 | 0.85 | **0.97** | 0.81 | **0.98** | 0.99 |
| **SMVAE** | 0.72 | **0.99** | 0.97 | **0.97** | 0.74 | **0.82** | 0.77 | **0.90** | 0.94 | 0.74 | 0.95 | **0.99** |
| **SMVGON** | 0.75 | **0.99** | **0.98** | **0.97** | **0.75** | **0.82** | 0.78 | **0.90** | 0.93 | 0.75 | 0.95 | **0.99** |

most of the subsets, though it is lower than MMVAE for single modalities. This indicates that approximations that minimize the divergence from the surrogate posterior are more effective than computationally expensive approximations based on cross-generation. In addition, the performance of SMVGON is almost the same as that of SMVAE, which means that the surrogate posterior using VGON is sufficient as a joint inference model even without inference networks. See Appendix for the results of inference and generation when training with VGON alone. This result confirms that the performance of the surrogate posterior is sufficient for this dataset.

The results for generation in Table 2 are similar to those of the inference of the representation. It can be seen that the performance of the previous work on generating SVHN from subsets is greatly degraded. This is thought to be due to the difficulty in cross-generation to SVHN, which resulted in poor learning. On the other hand, the proposed method is able to generate from subsets with the same performance as from all modalities. This shows that the proposed method achieves our objective, i.e., to make inference from any modality as good as from all modalities.

Figure 1 shows the results of SVHN generation given different modalities for MoPoE-VAE, SM-VAE, and SMVGON. We can see that MoPoE-VAE fails to generate SVHN from both MNIST and Text modalities. On the other hand, SMVAE can generate corresponding and diverse SVHN images from each modality. Sutter et al. (2020) attempted this generation by introducing modality-specific latent variables, but these results suggest that our proposed SMVAE can perform cross-generation of SVHN with only a single latent variable. In addition, it can be confirmed that SMVGON can generate images with similar performance, indicating that our objective can be achieved without inference networks.

## 5.2 BIMODAL CELEBA

Table 3 shows the results of representation and generation for Bimodal CelebA. On this dataset, SMVAE performs better than the other methods. We believe this is due to the challenging nature of this dataset. Although this dataset is bimodal, it has a larger diversity of images and text compared to MNIST-SVHN-Text. Therefore, it becomes difficult to learn reconstructions and cross-generations from a single modality, which makes previous methods that rely on this to learn inference insuffi-

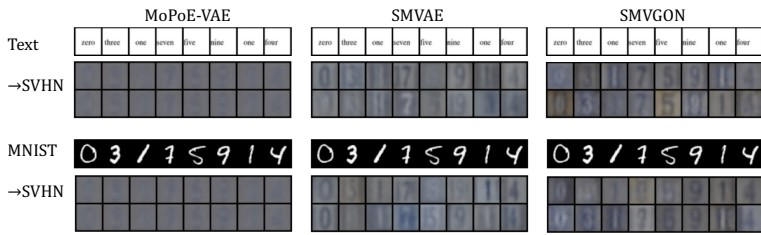

Figure 1: Cross-generation of SVHN on MoPoE, SMVAE, and SMVGON. Each row of SVHN in each cross-generation corresponds to a different sample from the inference model.

Table 3: Performance on the latent representation and modality generation in Bimodal CelebA. I and T respectively stand for Image and Text. We report the mean average precision across all modalities.

| | LATENT REPRESENTATION | | | GENERATION | | | |
| | | | | I | | T | |
| MODEL | I | T | I+T | T | I+T | I | I+T |
|---|---|---|---|---|---|---|---|
| MVAE | 0.48 | 0.39 | 0.54 | 0.30 | 0.51 | 0.25 | **0.50** |
| MMVAE | 0.51 | 0.48 | 0.50 | 0.45 | 0.45 | 0.34 | 0.34 |
| MOPOE-VAE | 0.51 | 0.43 | 0.53 | 0.42 | 0.50 | 0.32 | 0.40 |
| **SMVAE** | **0.56** | **0.56** | **0.59** | 0.46 | **0.53** | **0.36** | 0.50 |
| **SMVGON** | 0.50 | **0.56** | 0.52 | **0.47** | 0.45 | 0.32 | 0.45 |

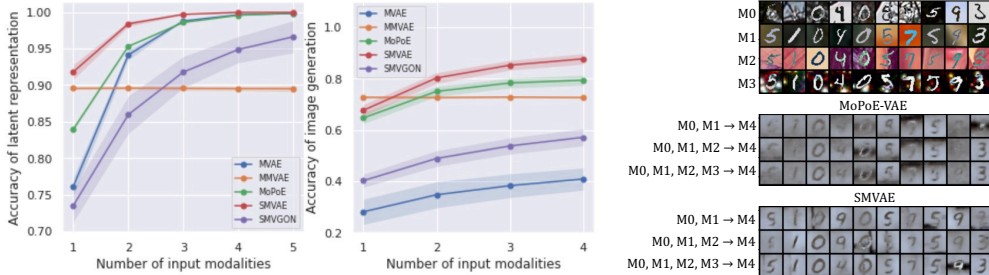

Figure 2: Performance on the PolyMNIST dataset. Left: Performance of the inferred latent representation given a set of modalities for each number. The markers are the means of the three runs, and the error bands are their standard deviations. Center: Performance of generating modalities given a set of modalities corresponding to each number. Right: Conditional generation of PolyMNIST.The top panel shows examples of modalities M0 to M3 (we refer to the five different modalities as M0 to M4), and the middle and bottom panels show the generation of M4 conditioned on the respective modality sets by MoPoE-VAE and SMVAE. Images in each column belong to the same example.

cient. Our proposed method, on the other hand, learns the surrogate joint posterior by reconstructing all modalities, and approximates the inferences from arbitrary subsets to be close to it. Therefore, our method is less sensitive to the difficulty of cross-generation. On the other hand, SMVGON has not reached the same level of performance as SMVAE. This may be due to the difficulty of the multimodal dataset, which makes VGON's inference insufficient. For the training results of VGON alone on this dataset, please refer to Appendix C. For qualitative generation results, see Appendix E.

## 5.3 POLYMNIST

Figure 2 shows the results of PolyMNIST. First, we can see that SMVAE outperforms the existing methods in both representation and generation. Also, the generation results show that SMVAE is able to generate corresponding numbers more clearly than MoPoE-VAE. This result confirms that SMVAE is a scalable model not only in terms of computational cost but also in terms of performance when the number of modalities increases. On the other hand, the performance of SMVGON is lower than that of other models in both generation and representation. This might be because, as in the case of bimodal CelebA, inference by VGON becomes difficult as the number of modalities increases.

## 5.4 CONCLUSION

In this study, we proposed SMVAE, which approximates inference from subsets by explicitly minimizing the divergence from the surrogate posterior distribution. The model is scalable with respect to the number of modalities and shows high performance in inference and generation, especially in the case where cross-generation is difficult. Furthermore, to reduce the parameters of the inference model, we introduced SMVGON using VGON, which can learn inference without inference networks. The remaining task of this study is to find a way to introduce a surrogate posterior distribution that will result in good inference with low memory and computational costs. VGON performed well on MNIST-SVHN-Text, but did not perform well on challenging problem settings such as Bimodal CelebA and PolyMNIST. In the future, we would like to address this issue and pursue effective methods for a wide range of multimodal datasets.

# 6 ETHICS STATEMENT

This work is a state-of-the-art method for multimodal VAEs, which will have an impact on the machine learning domain due to its scalability. Our method can cross-generate from one modality to another corresponding modality; therefore, we have to be aware that it can be used to generate non-existent facial images, text, spoken voices, etc. In addition, since the attributes in the CelebA dataset used for training in this study have biases (Prabhu et al., 2019), it is necessary to remove these biases when training the proposed method in practice.

# 7 REPRODUCIBILITY STATEMENT

All codes and datasets necessary to reproduce the experiments in this paper will be available after the paper review is concluded. The hyperparameters and other details in the training were written at the beginning of Section 5, and the architecture of the networks is the same as in Sutter et al. (2020).

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

## A  COMPARISON BETWEEN SMVAE AND JMVAE WHEN THE NUMBER OF MODALITIES IS 2

When the number of modalities is two, the objective function of JMVAE (Suzuki et al., 2016) is as follows:

$$\mathcal{L}(\Theta, \Lambda; X) - \beta D_{\mathrm{KL}}\left(q_\Lambda(\mathbf{z}|X)||q_{\phi_1}\left(\mathbf{z}|\mathbf{x}_1\right)\right) - \beta D_{\mathrm{KL}}\left(q_\Lambda(\mathbf{z}|X)||q_{\phi_2}\left(\mathbf{z}|\mathbf{x}_2\right)\right). \tag{13}$$

Compared to Equation 11, the difference is that the direction of KL divergence is reversed and aggregation by PoE is not used. Table 4 shows the results of comparing SMVAE and JMVAE on the Bimodal CelebA dataset. The performance of the representations is almost the same, but the generation results are better for SMVAE. This may indicate that the KL divergence direction of SMVAE is better for cross-generation.

## B  COMPARISON OF VAE AND VGON IN MNIST-SVHN-TEXT

Table 5 shows the inference and generation performance of VAE and VGON for multimodal inputs. It can be seen that VGON can generally learn inference models equivalent to VAE without inference networks, although its performance is slightly lower than that of VAE.

## C  COMPARISON OF VAE AND VGON IN BIMODAL CELEBA

Table 6 shows the comparison between VAE and VGON in Bimodal CelebA. The results confirm that the performance of VGON is lower than that of VAE in all tasks.

## D  EFFECTIVENESS OF END-TO-END LEARNING IN SMVAE

Figure 3 compares the results of end-to-end learning in SMVAE, i.e., optimizing Equation 11, and the results of two steps, i.e., first optimizing surrogate inference and then minimizing KL divergence with inference from arbitrary modalities. The dataset is PolyMNIST; in two stages, 200 epochs were trained in the first stage and 100 epochs in the second stage. The results show that the end-to-end method gives better results. This may be due to the fact that the two distributions are more easily approximated by training them simultaneously.

## E  CONDITIONAL GENERATION ON BIMODAL CELEBA

Figure 4 shows the results of conditional generation of Bimodal CelebA given a text modality. We used SMVAE as the model.

Table 4: Performance on the latent representation and modality generation in Bimodal CelebA.

| MODEL | Latent representation | | | Generation | | | |
|---|---|---|---|---|---|---|---|
| | | | | I | | T | |
| | I | T | I+T | T | I+T | I | I+T |
| JMVAE | 0.56 | 0.56 | **0.59** | 0.42 | **0.54** | 0.30 | 0.49 |
| **SMVAE** | **0.58** | **0.56** | 0.59 | **0.46** | 0.53 | **0.36** | **0.50** |

Table 5: Classification accuracy on the latent representation and generated modalities in MNIST-SVHN-Text. M, S and T respectively stand for MNIST, SVHN, and Text, indicating that the representations are inferred from these modalities. M+S means both M and S. A→ B means that the corresponding column is the performance of generating B conditioned on A.

| | Latent representation | Generation | | |
| Model | M+S+T | M+S+T → M | M+S+T→S | M+S+T→T |
|---|---|---|---|---|
| VAE | 0.99 | 0.96 | 0.87 | 0.99 |
| VGON | 0.99 | 0.91 | 0.85 | 0.99 |

Table 6: Classification accuracy on the latent representation and generated modalities in Bimodal CelebA.

| | Latent representation | Generation | |
| Model | I+T | I+T → I | I+T→T |
|---|---|---|---|
| VAE | 0.59 | 0.97 | 0.50 |
| VGON | 0.57 | 0.92 | 0.45 |

## F    RANDOM IMAGE GENERATION

Figure 5 shows the results of generating all modalities from random noise in the latent space for each dataset. We used SMVAE as the model. Each sample was generally generated correspondingly across modalities, but the text for Bimodal CelebA was not generated well.

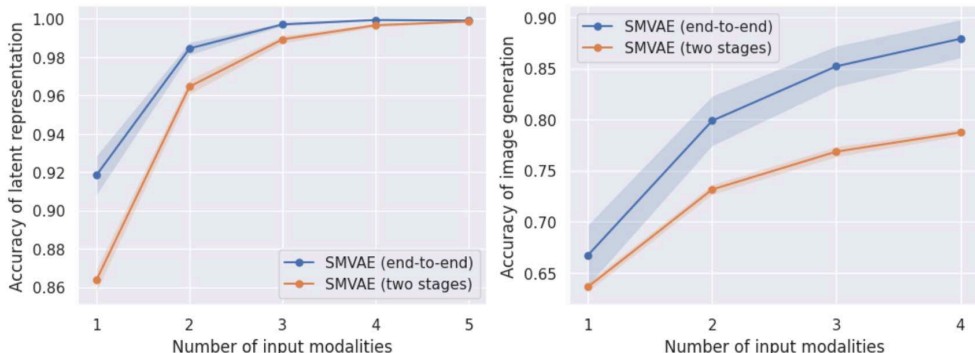

Figure 3: Performance on the PolyMNIST dataset. Left: Performance of the inferred latent representation given a set of modalities for each number as input.Right: Performance of generating modalities given a set of modalities corresponding to each number.

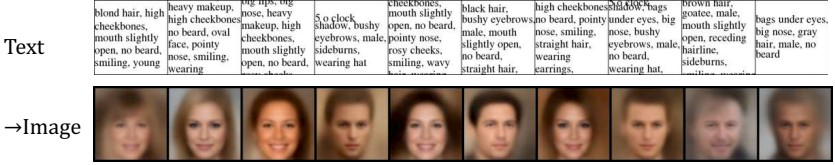

Figure 4: Conditional generation of Bimodal CelebA on SMVAE.

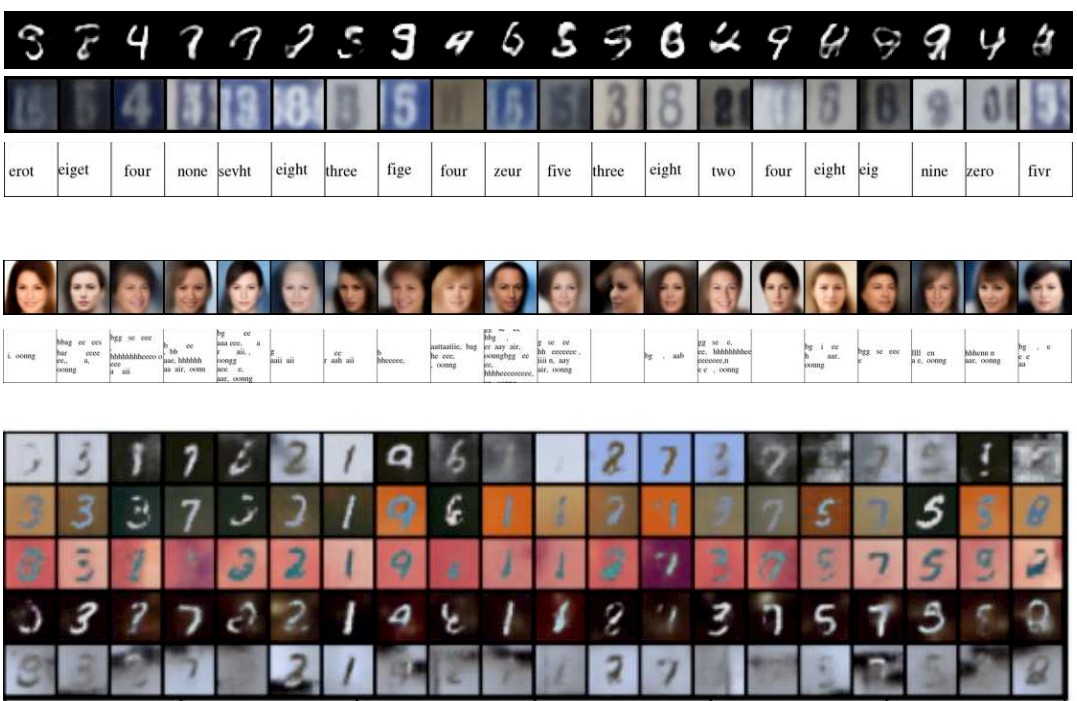

Figure 5: The results of generating all modalities from random noise in the latent space on each dataset. Top: MNIST-SVHN-Text (from top to bottom: MNIST, SVHN, Text). Center: Bimodal CelebA (Image ,Text). Bottom: PolyMNIST (M0, M1, M2, M3, M4). Each column of each dataset corresponds to the same random noise.

