# OpenReview forum: "Scalable multimodal variational autoencoders with surrogate joint posterior"
_ICLR.cc/2022/Conference — ICLR 2022 Submitted_

### Official Review · Reviewer_FDzr · 2021-11-02

**Correctness:** 4
**Technical Novelty And Significance:** 3
**Empirical Novelty And Significance:** 2
**Recommendation:** 5
**Confidence:** 4

**Main Review:**

### Strengths
- Interesting approach to scalable, multimodal VAEs. It seems to be a reasonable trade-off between computational complexity and memory-constraints.
- Disregarding some typos and missing information (see minor comments), clear description of the proposed method

### Weaknesses
- For empirical evaluation, quantitative results on the quality of generated samples are missing as well as classification results of randomly generated samples (see Shi et al, 2019). Also, there are no results reported with respect to test set log-likelihoods. Previous work on multimodal VAEs showed that there is a trade-off between quality of generated samples and their coherence (see Sutter et al, 2021). Hence, without any quantitative results regarding the quality of samples, the assessment of a model’s performance is somehow not complete in my opinion.
- There are multiple typos in this work (e.g. eq. 10)

### Questions
- In Sutter et al, 2020, the authors propose a similar concept as yours. Using the Jensen-Shannon-divergence, the unimodal posterior approximation are forced to be similar to the joint posterior approximation. Could you share your thoughts on the similarity between the two approaches?
- Shouldn’t the results in Table 1 for the latent representation and the results in Table 2 (last row, 2 rows at the bottom) be the same numbers?
- In section 3.2, what do you mean by ‘it is unlikely that the performance of inference and generation will be degraded..’? Unlikely is not a very technical term. Therefore, it is difficult to assess this statement in my opinion.

### Minor Comments
- I had difficulties finding the definition of the surrogate joint posterior. I think it is defined as Gaussian distribution as well. I could only find it for the VGON case, but not for the VAE case of the proposed method. Is this correct?
- I would appreciate to see some qualitative results for the CelebA experiment as well.

**Summary Of The Paper:**

The authors propose a new approach to the class of scalable, multimodal VAEs. They propose to use a surrogate joint posterior which, using an additional sum of KL-divergence terms, helps keeping the posterior approximation of subsets of modalities similar to the joint posterior approximation.
The proposed method is evaluated on two different datasets, MNIST-SVHN-Text and bimodal CelebA where the proposed method shows good results.

**Summary Of The Review:**

This is an interesting approach to scalable, multimodal learning with promising results regarding the learned latent representations and coherence of generated samples. Nevertheless, quantitative results on the quality of generated samples (including random samples) are absolutely necessary for multimodal VAEs in my opinion. Therefore, I can only give a weak reject at this point.

---

> ### Author Response · Authors · 2021-11-23
> **Reply to reviewer FDzr**
>
> > Evaluation of sample quality
>
> Thank you for your valuable comments. Since this research aims to ensure that inference from any given modality set is performed properly (in other words, there is no significant performance loss in any of the modality sets), we focused on evaluating representation and cross-modal rather than random generation. However, it is essential to check random generation in multimodal VAEs, so we have included qualitative results of random sample generation on all datasets in the Appendix of this paper.
>
> > Quantitative evaluation of log-likelihood
>
> Conventional multimodal VAEs, such as MMVAE and MoPoE-VAE, correspond to being good inference models when their ELBO (log-likelihood) is maximized. However, this is not the case for the SMVAE proposed in this study because the log-likelihood corresponding to the first term in Equation 11 only indicates the goodness of the surrogate inference model and is not directly related to the performance of inference from arbitrary modalities.
> Also, since the objective function of SMVAE, Equation 11, is no longer a valid variational lower bound, it is not essential to evaluate this equation itself and compare it with MoPoE-VAE and other multimodal VAEs. Therefore, in this study, we did not use the log-likelihood but rather the performance of representation and generation.
>
> > Differences with MMJSD
>
> We believe that there are two differences with MMJSD.
> First, MMJSD only considers unimodal inference in KL divergence, whereas in our work, we approximate joint inference with any combination of unimodal, which allows us to obtain inference from any set of modalities.
> The second is the design of the joint inference. In MMJSD, they introduce a dynamic prior that combines all unimodal inferences and consider it as a joint inference. However, there is no guarantee that this dynamic prior approximates the true posterior as the surrogate posterior of SMVAE (the inference model of ELBO for MMJSD and the dynamic prior are different forms). Another difference is that MMJSD uses the same network for joint inference and unimodal inference. We have added these explanations in the section of related work in the revised paper.
>
> > What do you mean by 'it is unlikely that the performance of inference and generation will be degraded...'? ?
>
> As you pointed out, this was an inappropriate explanation. What we were trying to explain is that existing multimodal VAEs such as MMVAE and MoPoE-VAE rely on the performance of reconstruction or cross-generation to approximate inference from an arbitrary set of modalities. Therefore, if reconstruction or cross-generation is difficult to obtain in some cases, the inference included in the process will fail to approximate, resulting in poor inference performance for that modality and cross-generation via that inference. On the other hand, the proposed method does not rely on such reconstruction or cross-generation for approximation but instead minimizes the divergence between surrogate posterior in the latent space. This means that the proposed method is less prone to the degradation of inference that can occur with such reconstruction and cross-generation based methods. We have replaced the text with this explanation.
>
> > The definition of the surrogate joint posterior
>
> Thank you for pointing this out. Yes, you are correct. We have added an explanation of this after Equation 10 in the revised paper.
>
> > Qualitative results for the CelebA experiment
>
> We have included the results of conditioning generation for Bimodal CelebA by SMVAE in Appendix.

---

### Official Review · Reviewer_yRW1 · 2021-11-02

**Correctness:** 3
**Technical Novelty And Significance:** 3
**Empirical Novelty And Significance:** 2
**Recommendation:** 6
**Confidence:** 3

**Main Review:**

Strengths
-------------
- The authors do a good job of explaining the works that their paper builds upon.
- Through the experiments, the authors are able to confirm that their objective outperforms cross-generation-based objectives.

Weaknesses
------------------
- It is not clear why the ELBO and the KL-divergence (between the all-modality posterior and subset-modality posterior) need to be optimized simultaneously. A more intuitive approach would be to learn the all-modality posterior first by optimizing the ELBO and then optimize the KL-divergence term.
- The objective optimized in this paper appears to be a much weaker lower bound to the log-likelihood (due to the presence of the negative KL-divergence).
- The authors do not do a very good job of motivating the problem of learning representations when cross-modal generation is difficult. They should have conducted a few experiments where cross modal generation is difficult to ascertain that the representations learnt on these datasets by the baselines are bad.

**Summary Of The Paper:**

In this paper, the authors propose a model for learning joint representations for multimodal data that allows inference when a few of the modalities are missing. Previous approaches addressed the missing modality problem by treating the inference network for each modality as an expert and combining the information across these experts. The proposed model claims to be useful in the following scenarios:
1) When cross-generation is difficult (and hence, MoPoE-VAE fails).
2) When the number of modalities is large.

The authors achieve this by learning a surrogate posterior conditioned on all the modalities while simultaneously optimizing the KL-Divergence between the surrogate posterior and a subset-conditioned posterior. The surrogate posterior is learnt by optimizing a variational lower bound to the log-likelihood objective.
The authors evaluate the representations learnt by the model on secondary tasks and report improved performance. The generated modalities also perform well on secondary tasks.


**Summary Of The Review:**

Overall, the paper does a reasonable job of explaining the background for multimodal retrieval as well as conducting experiments to demonstrate their superior performance. However, the motivation for their work could be improved. The motivation for choosing the specific objective should also have been discussed in detail.

---

> ### Author Response · Authors · 2021-11-23
> **Reply to reviewer yRW1**
>
> > Why the ELBO and the KL-divergence need to be optimized simultaneously
>
> There are two reasons for this. The first reason is that learning the ELBO and the KL-divergence separately takes more time to optimize each of them. Learning ELBO and KL divergence simultaneously has been done in previous multimodal studies such as JMVAE, and we followed this method in this study. Secondly, empirically, learning them simultaneously gives a better approximation of the inference than learning them separately, and improves the performance of generation and inference from arbitrary modalities. In the Appendix of this paper, we compare the performance of end-to-end training with that of two-step training. The objective optimized in this paper appears to be to improve performance.
>
> > The objective optimized in this paper appears to be a much weaker lower bound to the log-likelihood
>
> As you pointed out, from the surrogate posterior's point of view, the proposed method's objective function seems only to deteriorate the approximation performance. However, what we actually want to optimize is PoE-aggregated inference from arbitrary modalities, and as mentioned above, end-to-end learning will improve the performance of those inferences.  This is probably because simultaneous optimization makes it easier to get all PoE-aggregated inferences closer together via surrogate posteriors.
>
> > The authors do not do a very good job of motivating the problem of learning representations when cross-modal generation is difficult.
>
> As you pointed out, We should have explained this issue more clearly and in more detail. We have added a few additions to Section 3.1 to explain this issue a bit more. We have added some more explanations of this issue in Section 3.1.

---

> > ### Comment · Reviewer_yRW1 · 2021-11-30
> > **Reply to authors**
> >
> > In JMVAE, the KL divergence term is a part of the ELBO. Hence, the statement "Learning ELBO and KL divergence simultaneously has been done in previous multimodal studies such as JMVAE" is incorrect.
> >
> > In your case, the KL-divergence (between the all-modality posterior and subset-modality posterior) is an additional term that weakens the ELBO. The resultant all-modality posterior learnt by the model will be worse than the one learnt if ELBO had been optimized alone. I don't see any reason (other than time required for training) why you would want to worsen the all-modality posterior q_\Lambda (z|x)
> >
> > I will keep my rating as it is.

---

> > > ### Author Response · Authors · 2021-12-01
> > > **Reply to reviewer yRW1**
> > >
> > > We are grateful for your reply. However, we would disagree with your statement that the JMVAE's KL divergence terms are included in the ELBO. According to the original JMVAE paper [1], the objective function (Equation (4)) is the ELBO with multimodal input *plus* the KL divergence terms between the all-modality posterior and the unimodal posterior. In the JMVAE paper, the model based on this objective function is called "JMVAE-KL", but in many papers today it is called just "JMVAE". Therefore, we believe your statement that "In JMVAE, KL divergence term is a part of the ELBO" is incorrect (of course, the KL divergence between the all-modality posterior and *prior* is included in the ELBO, but this is not about unimodal posterior we mentioned.).
> > >
> > > Also, we would like to reply to your second point. Since the all-modality posterior is Gaussian, no amount of ELBO optimization can completely approximate the true posterior. In addition, it is also difficult to completely approximate the all-modality posterior for all inferences from any set of modalities. Therefore, we believe that it is empirically better to balance these factors and optimize the all-modality posterior by considering both the true posterior and inferences from arbitrary modalities from the beginning of the learning process.
> > >
> > > [1] Joint Multimodal Learning with Deep Generative Models

---

### Official Review · Reviewer_MNSw · 2021-11-03

**Correctness:** 3
**Technical Novelty And Significance:** 4
**Empirical Novelty And Significance:** 3
**Recommendation:** 8
**Confidence:** 2

**Main Review:**

Good:
+ Linear increase in the number of parameters with growing number of modalities is a strong feature of the presented model
+ The computational complexity of evaluating the loss function is substantially reduced, which plays an important role with a large number of modalities
+ The motivation and connection to the previous work is clearly indicated, and the paper is easy to follow

Concerns:
1. Only two datasets are considered, which have 2 and 3 modalities. Since the scalability of the computational cost of SMVAE with the increasing number of modalities is one of the main motivations for the suggested method, benchmarking it on more modalities seems important. The paper follows the experiments conducted in MoPoE-VAE study, with the exception of PolyMNIST of 5 modalities. Conducting this experiment would be crucial for supporting the main claims of the paper and completing the benchmark against MoPoE-VAE.

2. The evaluations for several experiments do not have the same results as reported in the MoPoE-VAE paper. this affects especially Table 2, where the results of MoPoE-VAE are often a lot worse than in the original publication(also, Table 2). As far as I can see, the results of SMVGON in table 3 all lie within the reported confidence intervalls

3. The cross-generation improvement is one of the main points of the paper. However it is not really supported by the presented MNIST-SVHN-Text experiments. Out of 9 “cross-generation” experiments in Table 3, where the modality of interest is not part of the input (e.g. generating text from MNIST and SVNH), only 4 result in SMVAE outperforming MoPoE-VAE and MMVAE (as indicated by bold coloring, see minor concern below)

4. Very few qualitative examples of MNIST-SVHN-Text, and none of Bimodal Celeba


Small concerns:
-almost all tables miss proper description and caption. e.g., in Table 3 it is guesswork to figure out which of the header rows mean what.
- The result tables mark several results bold as the best, even though they only differ in the third digit. It is unclear whether those differences are significant
- Some previous multi-modal VAE models are relevant but are not part of the literature review: PoE-based SVAE model arXiv:2101.07240, which is a generalisation of MVAE and has a quadratic increase in parameters with increasing number of modalities, and VAEVAE arXiv:1912.05075 that has architectural resemblance also using the surrogate joint posterior (or a joint encoder, as introduced in JMVAE study).

Concern 1 is for me the most important one and it is unclear to me why this experiment has not been conducted. Otherwise the authors should clarify the result differences in Concern 2.

**Summary Of The Paper:**

SMVAE aims to solve the challenge of combining multiple modalities into a coherent latent space using VAE framework. SMVAE is based on MoPoE-VAE model, the main architectural change is replacing the MoE rule with minimising the KL-divergence between the latent space distribution generated by the PoE combined experts and the joint encoder.


(The reviewer is sorry for the slight delay for submission as he is still on vacation and marked november third as deadline for submission. In the interest of time,the following is just a short summary of my notes)

**Summary Of The Review:**

The paper includes a novel formulation of  multi-modal VAE learning, but the experimental section does not support the main claim of scalability of training and the results of the strongest reported baseline have large differences from the ones reported in the original publication. If these problems are addressed, I can see raising my score.

---

> ### Author Response · Authors · 2021-11-23
> **Reply to reviewer MNSw**
>
> > Only two datasets are considered, which have 2 and 3 modalities.
>
> We have added an experiment with PolyMNIST with five modalities. This experiment confirms that SMVAE performs better in generation and inference than existing multimodal VAEs, including MoPoE-VAE. On the other hand, the performance of SMVGON was significantly worse. This is probably because a one-step inference model like VGON does not provide sufficient inference for complex input that includes many modalities. These results can be found in Section 5.3 and Figure 2.
>
> > The evaluations for several experiments do not have the same results as reported in the MoPoE-VAE paper.
>
> Although we tried to implement as close to the original work as possible, we admit that we could not reproduce the performance of some experiments sufficiently. This may be due to differences in the implementation environment. On the other hand, for PolyMNIST, we could reproduce results close to those in the original paper. We would like to emphasize that all models were tested in the same environment, with the same architecture and hyperparameters, and we are comparing them fairly.
>
> > The cross-generation improvement is one of the main points of the paper. However it is not really supported by the presented MNIST-SVHN-Text experiments .
>
> We argue that existing multimodal VAEs rely on cross-generation to approximate inferences. If the cross-generations are difficult to learn (e.g., if the mapping between modalities is difficult), the inferences may not be sufficiently learned, resulting in significantly inferior inferences and generation from some modalities. The proposed method does not rely on such cross-generation to approximate inference but instead minimizes the divergence in the latent space between surrogate posterior. By doing so, we aim to improve the performance of modalities whose inference and generation have been significantly degraded by the above issues. Therefore, our main goal is to keep the performance of inference and generation from any given modality similar to that of other modalities without significant degradation. In this context, the fact that MNIST-SVHN-Text improves the generation performance for SVHN, which has degraded significantly, corresponds to our goal. In addition, PolyMNIST experiments show the best average evaluation of inference and generation from a certain number of modalities. This indicates that our method improves inference and generation, which is inferior to previous methods.
>
> > Very few qualitative examples of MNIST-SVHN-Text, and none of Bimodal Celeba
>
> We have added qualitative evaluation to the Appendix. We show the conditional generation of Bimodal CelebA and the random generation on all datasets.
>
> > The result tables mark several results bold as the best, even though they only differ in the third digit.
>
> As you pointed out, the difference in the third digit is not a big difference. Therefore, all the results in the table have been corrected with two significant digits.
>
> > Some previous multi-modal VAE models are relevant but are not part of the literature review
>
> Thank you for presenting the related studies. We have explained them in Section 4 and clarified the differences with the proposed method.

---

> > ### Comment · Reviewer_MNSw · 2021-11-29
> > **Thanks for clarifications**
> >
> > Thanks for providing the updated tables and experimental results. I have updated my score. I think the approach has merit and performs reasonably well, when compared to previous work. I am not sure this technique will survive the test of time, but I am reasonably confident that it will have some impact - at least the VAE part.
> >
> > I did not check this, but if you still report your own results of MoPoE-VAE, then state that you were not able to reproduce the same performance as reported in the original paper.

---

### Official Review · Reviewer_yNag · 2021-11-05

**Correctness:** 2
**Technical Novelty And Significance:** 2
**Empirical Novelty And Significance:** 3
**Recommendation:** 3
**Confidence:** 3

**Main Review:**

Perhaps it is because I am not an expert in multi-modal VAEs, but I found this paper quite hard to read.  It presents to the reader a series of models, variational formulations, and ELBOs, and I had a hard time tracking the relationships between each.  My primary critique then is on the paper's presentation of its innovation, which takes form in Equations 10 and 11.  I really wanted to see a derivation that connects equations 10 and 11.  Since there are variational models on both sides of Equation 10, it's not clear to me how the (generative) model joint $P(X, Z)$ arises in the ELBO presented in Equation 11.  Relatedly, I found the paper's claims about the proposed framework, such as "it is unlikely that the performance of inference and generation will be degraded by the difficulty of cross-generation" (p 6) to be hard to agree with or verify as these concepts are not defined with any rigor.

**Summary Of The Paper:**

This paper addresses the problem of formulating a well-justified ELBO for multi-modal VAEs.  The paper gives a thorough recounting of previous attempts at combining and training the per-mode posterior (approximations).  Previous workarounds involve, for instance, subsampling subsets.  This paper proposes approximating the joint posterior with a variational model and then adding the KLD to this variational posterior as an extra term on the ELBO, analogous with but an approximation to the KLD between the approximate and true posteriors that controls the gap in the ELBO.  The experiments report classification performance on MNIST + SVHN + Text and bi-modal CelebA.

**Summary Of The Review:**

The paper's presentation---specifically, the lack of a derivation that connects Eqs 10 and 11---makes it very hard for me to assess the paper's contributions.  Thus, I think the paper needs a major revision to improve clarity.

---

> ### Author Response · Authors · 2021-11-23
> **Reply to reviewer yNag**
>
> > A derivation that connects equations 10 and 11
>
> As you pointed out, the relationship between these two equations was not fully explained. Therefore, we have added this detailed explanation in the revised paper. For the reasons for using equation 11 as the objective function, please refer to my response to reviewer yRW1 and the revised paper.
>
> > "it is unlikely that the performance of inference and generation will be degraded by the difficulty of cross-generation" (p 6) to be hard to agree with or verify as these concepts are not defined with any rigor.
>
> Thank you for pointing this out. This was an inappropriate explanation. We have corrected it in the revised paper. Also, please refer to our response to reviewer FDzr for the point we wanted to make in this sentence.

---

### Author Response · Authors · 2021-11-23
**Reply to all reviewers**

Dear reviewers,

Thank you all for the constructive feedback. Based on your review, We have made the following revisions to the paper.
- We have added the PolyMNIST experiment. Due to page limitations, We have removed Figure 1 and moved Table 1 to Appendix.
- The results of conditional generation of Bimodal CelebA and random generation of all modalities on all datasets are included in Appendix.
- We have explained in detail the process of deriving Equation 11 from Equation 10. We also explain why we optimize Equation 11 end-to-end instead of a two-stage optimization and show the difference in performance between the two cases in Appendix.
- We have changed the significant digits in the results to two according to previous studies.
- In the related studies section, we have added a description of the studies related to this one that the reviewer MNSw pointed out and explained how they differ from our model.

In addition to the feedbacks above, we have corrected some spelling and notation errors. In addition, some redundant sentences have been deleted or rewritten in order to adjust the number of pages.

---

### Decision · Program_Chairs · 2022-01-20

**Decision:**

Reject

**Comment:**

PAPER: This paper proposes a method to learn joint representations from potentially missing data when (1) cross-generation may be difficult, and/or (2) with large number of modalities. This is achieved by minimizing the divergence between a surrogate joint posterior and inferences from arbitrary subsets.
DISCUSSION: The reviews and discussion brought many relevant issues and concerns. The authors submitted a revised version that improved the clarity of the paper and added an important experiment with PolyMNIST. In their responses, authors also addressed some misunderstanding about JMVAE-KL. The comparison with a relatively similar work, from Sutter et al., 2020, was only mentioned in the related work, with no direct comparisons. Also, the authors did not directly address the issue of studying tradeoffs between quality of generated samples and their coherences. It should also be noted that the advantage of the proposed SMVAE is marginal when the number of modalities increases, for the latent representation experiments on PolyMNIST.
SUMMARY: Enthusiasm for this paper was not unanimous. The reviewers brought some concerns about its differentiation with priori work, such as Sutton et al., 2020, and about a more detailed analysis of the tradeoffs. While the clarity of the paper improved during the revision, a good number of issues remained. I am leaning towards rejection.